# Multi-Task Structural Learning using Local Task Similarity induced Neuron Creation and Removal

## Abstract

Multi-task learning has the potential to improve generalization by maximizing positive transfer between tasks while reducing task interference. Fully achieving this potential is hindered by manually designed architectures that remain static throughout training. On the contrary, learning in the brain occurs through structural changes that are in tandem with changes in synaptic strength. Thus, we propose *Multi-Task Structural Learning (MTSL)* that simultaneously learns the multi-task architecture and its parameters. MTSL begins with an identical single task network for each task and alternates between a task learning phase and a structural learning phase. In the task learning phase, each network specializes in the corresponding task. In each of the structural learning phases, starting from the earliest layer, locally similar task layers first transfer their knowledge to a newly created group layer before they are removed. MTSL then uses the group layer in place of the corresponding removed task layers and moves on to the next layers. Our empirical results show that MTSL achieves competitive generalization with various baselines and improves robustness to out-of-distribution data. [1]

## 1 Introduction

Artificial Neural Networks (ANNs) have exhibited strong performance in various tasks essential for scene understanding. Single-Task Learning (STL) (Yu et al., 2021; Wang et al., 2020b; Orsic et al., 2019) has been largely at the center of this exhibit driven by custom task-specific improvements. Despite these improvements, using single task networks for the multiple tasks required for scene understanding comes with notable problems such as a linear increase in computational cost and a lack of inter-task communication.

Multi-Task Learning (MTL), on the other hand, with the aid of shared layers provides favorable benefits over STL such as improved inference efficiency and positive information transfer between tasks. However, a notable drawback of sharing layers is task interference. Existing works have attempted to alleviate task interference by modifying the architecture (Kanakis et al., 2020; Liu et al., 2019), determining which tasks to group together using a similarity notion (Standley et al., 2020; Fifty et al., 2021; Vandenhende et al., 2020), balancing task loss functions (Kendall et al., 2018; Liu et al., 2019; Yu et al., 2020; Lin et al., 2019), or learning the architecture (Guo et al., 2020; Lu et al., 2017). Although these methods have shown promise, progress can be made by drawing inspiration from the brain, which is the only known intelligent system that excels in multi-task learning. The inner mechanisms of the brain, although not fully understood, can guide research in ANNs through simplified notions. Neuron creation and neuron removal (Maile et al., 2022) are simplified notions that can aid in the automated design of Multi-Task Networks (MTNs).

**Neuron removal** presents the opportunity to start from a dense set of neurons and move toward a sparse set of neurons. In the early stages of brain development, neural circuits consist of excess neurons and connections that provide a rich information pipeline (Maile et al., 2022). This pipeline allows neural circuits to learn specialized functions while undergoing neuron removal and synaptic pruning (Riccomagno & Kolodkin, 2015). Thereby, moving from a dense architecture consisting of multiple single-task networks to a sparse multi-task architecture could be beneficial.

---

[1]Code will be made available after acceptance.

**Neuron creation** is an open-ended operation due to the difficulty involved in deciding where, how, and when to create neurons (Evci et al., 2022). In the brain, local communication between neurons is an important part of learning. Learning rules that modulate synaptic strength are local in nature (Kudithipudi et al., 2022) and local neural activity could be responsible for the creation of neurons (Luhmann et al., 2016) and also neuron removal (Faust et al., 2021). We explore local task similarity to drive neuron creation and removal, which together could improve learning.

**Structural learning** pertains to the learning of the architecture and its parameters simultaneously (Maile et al., 2022). The neural circuitry in the brain changes even during adulthood, undergoing morphological changes induced by structural plasticity (Kudithipudi et al., 2022). Evidently, learning in the brain does not involve static architecture creation followed by modulation of synaptic strengths. Instead, architecture changes occur in tandem with changes in synaptic strength. Thus, utilizing structural learning with strategic neural operations could mitigate the effects of task interference and promote generalization in MTL.

Therefore, we propose Multi-Task Structural Learning (MTSL) to simultaneously learn the multi-task architecture and its parameters. MTSL considers entire layers as computation units (Maile et al., 2022) and performs neuron creation and neuron removal on them. Inspired by the creation of a large number of neurons in the developmental stage of the brain, MTSL begins training by initializing each task with its own network. Similar to the brain, the excess layers of each task network provide a rich information flow to inform grouping decisions. Local task similarity is used to guide task learning through the alignment of task representations, and also to make decisions on grouping tasks. A positive decision to group tasks induces the creation of a group layer, and the associated task layers transfer their learned knowledge to the group layer before being removed. Finally, a few epochs of fine-tuning result in a learned MTN which persists the learned parameters for inference.

**Contributions**. (i) We propose a structural learning algorithm for multi-task learning based on aligning local task representations, grouping similar task layers, transferring information from grouped task layers to a new group layer, and removing the concerned task layers. (ii) We compare against various state-of-the-art methods and show that MTSL shows improved generalization without the need to retrain. (iii) We show that MTSL improves the robustness to natural corruptions. (iv). We present an ablation on the various components of MTSL and show its utility.

## 2 RELATED WORKS

Although different lines of work, such as architecture modifications (Liu et al., 2019; Kanakis et al., 2020; Misra et al., 2016), task grouping (Standley et al., 2020; Fifty et al., 2021; Vandenhende et al., 2020), or task loss balancing (Kendall et al., 2018; Liu et al., 2019; Yu et al., 2020; Lin et al., 2019) address task interference, they use hand-designed architectures that could be suboptimal. A variety of works in the MTL literature propose methods to learn the architecture, and we categorize these works into two groups. One group of works considers learning architectures that are capable of dynamically changing their structure based on the input (Hazimeh et al., 2021; Ahn et al., 2019; Rosenbaum et al., 2018) while the other group learns the branching structure (Guo et al., 2020; Bruggemann et al., 2020; Lu et al., 2017; Zhang et al., 2022; 2021; Raychaudhuri et al., 2022).

**Input dependent dynamic architectures** draw inspiration from the brain and provide many benefits, including improved computational efficiency (Han et al., 2021). DSelect-k (Hazimeh et al., 2021) is a mixture of experts model that enables selecting a spare set of experts to infer an input sample. (Ahn et al., 2019) learn a selector network that learns to pick a subnetwork from a large estimator network based on input. In routing networks (Rosenbaum et al., 2018), task-dependent agents are trained using reinforcement learning to pick a path within a large network to infer an input. While these approaches aim to optimize networks or subnetworks to specialize for a certain distribution of input samples, MTSL aims to optimize a shared network for all tasks.

**The branching structure** of multi-task networks have been learned using different approaches. (Zhang et al., 2022) propose to estimate the accuracy of a branched multi-task network using two task networks with similar branching. They also suggest data structures and methods to ease branching decisions search in an arbitrary network similar to (Zhang et al., 2021). Raychaudhuri et al. (2022) propose two controller networks that predict the branching structure and the weights of the cross-task edges based on user preferred task importance and budget constraints. (Guo et al., 2020) start from a

dense search space where a child layer is connected to a number of parent layers. During learning, a distribution over parent nodes is learned with the aid of path sampling. At the end of training, a valid network path is picked and using neuron removal, the neurons no longer a part of the valid path are removed. BMTAS (Bruggemann et al., 2020) takes a similar approach to (Guo et al., 2020) but additionally use a resource loss. On the contrary, MTSL involves progressive neuron removals at different intervals during training. The branching structure learning approaches discussed so far explicitly retrain the learned architecture, while MTSL avoids retraining and confirms with structural learning. Like MTSL, (Lu et al., 2017) also avoid retraining and use neuron creation where tasks are split into different branches starting from the output layer to the input layer using inter-task affinities defined based on task error margins. Unlike (Lu et al., 2017), MTSL starts from a dense set of neurons and moves towards a sparse architecture. Also, MTSL is designed to leverage both neuron creation and neuron removal. Adashare (Sun et al., 2020) learns task specific policies to determine which residual blocks to execute or skip, leading to residual blocks in the encoder specializing in a subset of tasks. Unlike (Guo et al., 2020; Zhang et al., 2022; 2021), BMTAS and MTSL, Adashare does not directly learn a branching structure and is specifically designed for ResNet.

## 3 MULTI-TASK STRUCTURAL LEARNING (MTSL)

The tasks to be learned together in an MTN bring in diverse information about the input scene. This diverse information can be leveraged to learn representations with improved generalization on all tasks. Design decisions such as which layers to share and where to branch tasks are complex due to their combinatorial nature. This complexity, along with the crucial role of these decisions in the interplay between positive transfer and task interference are reasons that could render manual architecture design suboptimal. Structural learning, on the other hand, learns the architecture along with its parameters, which is likely in line with how the brain learns.

MTSL uses two neural operators, namely neuron creation and neuron removal, to aid in structural learning. In early development, the brain has excess neurons that can provide a rich information pipeline for a pruned neural circuit to functionally specialize. Likewise, MTSL creates excess neurons by starting from a disparate network for each task. Through the progress of training, the corresponding task neurons in a layer pave the way for a specialized group neuron leading to a structural change. In the next sections, we present the finer details of the MTSL algorithm. In Section 3.1, we formalize the problem setup and establish the terminologies that we use in the rest of the paper. Following this section, we discuss how we align task representations in Section 3.2, how we create neurons in Section 3.3, removal of task neurons in Section 3.4 and the overall MTSL algorithm in Section 3.5.

### 3.1 PROBLEM SETUP

We consider the problem of structural learning where the MTN architecture and its parameters are learned simultaneously. Given the set of $\mathcal{T}$ tasks that each has its own single network with $L$ layers, our algorithm results in a single MTN capable of inferring all the $\mathcal{T}$ tasks accurately without the need for retraining. First, we establish the terminologies that are used in the rest of the paper. A **node** is a layer that connects one branch to another branch (or to a node), and a **branch** is a sequence of layers that follow a node. Initially, the first layer of each single task network is the task node, while the rest of the task network excluding the task head is called the task branch. Similarly, a group of tasks will have a group node and a group branch. A task node is of particular significance to our algorithm, as tasks can only be fused at the task node. Also, only task nodes that are connected to the same group branch or group node can be fused. At the start of the training, all task nodes are connected to the input image and can be fused. $T$ and $G$ denote a task and a group, respectively. Additionally, $F$ and $\mathcal{F}$ denote the output features of the task node and the group node, respectively. Figure 1 provides the overall schematic of our approach, where the leftmost column illustrates the initial state of our setup using the terminologies defined so far. In the following sections, we discuss the different components of our approach.

### 3.2 ALIGNING TASK SPECIFIC REPRESENTATIONS

As is evident from Figure 1, we begin training from single-task networks. Since the encoder of each task is initialized with ImageNet weights, there exists a correspondence between task nodes initially.

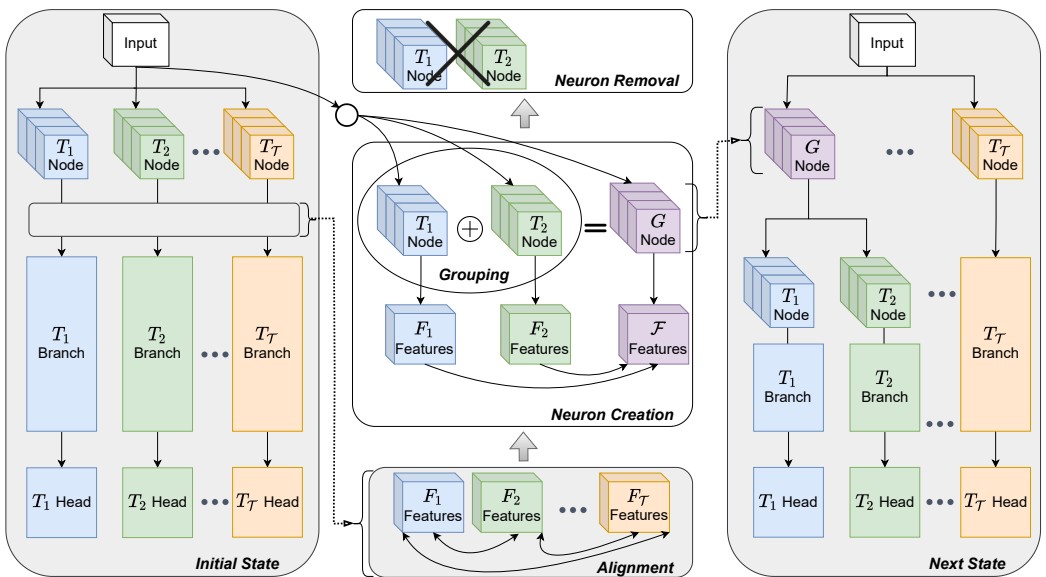

Figure 1: Schematic of the MTSL Algorithm. The grey regions (initial state, alignment, next state) are part of the task learning phase while the white regions (neuron creation, and neuron removal) are part of the structural learning phase. During training, our algorithm loops between alignment in the task learning phase followed by neuron creation and neuron removal in the structural learning phase leading to the next state (last column).

During training, task nodes would learn concepts that minimize particular task loss independent of other tasks. This independence likely breaks any correspondence between parameters mapped one-to-one between any two task nodes. This behavior comes from the permutation invariance of neural networks that leads to no guarantee on the order in which concepts are learned (Wang et al., 2020a). Therefore, MTSL aligns the concepts learned by the task nodes and locally increases their similarity using Centered Kernel Alignment (CKA) (Kornblith et al., 2019).

CKA is used to measure similarity between two feature representations and has been shown to provide meaningful similarity scores. During training, we introduce a CKA-based regularization term between task nodes branching from the same group node/branch (or the input). This regularization term, as shown in the alignment part of Figure 1, is included between all pairs of task node features (indicated by bi-directional arrows) and enforces the task representations to align by serving as an alignment constraint. We use the unbiased CKA estimator (Nguyen et al., 2021) to facilitate reliable estimates of CKA with small batch sizes used during training.

$$\mathcal{L} = \mathcal{L}_{MTL} + \lambda(1 - \mathcal{L}_{CKA}) \tag{1}$$

The overall loss that is used for training in the task learning phase of our algorithm is provided in Equation 1, where the first term ($\mathcal{L}_{MTL}$) represents the multi-task loss which is a weighted sum of all individual task losses. The second term represents the CKA regularization term ($\mathcal{L}_{CKA}$) that is included with a balancing factor $\lambda$ and a negative sign to maximize alignment between tasks.

### 3.3    CREATING GROUP NODES

The overall loss $\mathcal{L}$ used during the task learning phase (discussed in Section 3.2) leads tasks to learn similar features while also minimizing the concerned task loss. After the task learning phase, MTSL begins the structural learning phase to first leverage neuron creation. In the brain, local neuronal activity can affect the structure of the neural circuitry (Luhmann et al., 2016) and play a role in learning experiences (Kudithipudi et al., 2022). Taking cues from these notions of locality, we use CKA to gauge the similarity between task node features that represent the local activity of task neurons. These local task similarities are used to induce the creation of group nodes.

First, CKA between all pairs of task node features is calculated after which all possible groups of task nodes are listed. From these groups, a set of groups that maximize the total similarity is chosen and

the groups that satisfy a minimum similarity of $\gamma$ induce the creation of a group neuron. For instance, in Figure 1, we see that the groups picked are $[T_1, T_2]$ and $T_3$ assuming that the total number of tasks $\mathcal{T}$ is three. More details regarding the grouping algorithm are provided in the Appendix.

After the task nodes have been grouped based on local task similarity, for each group, a group node is created. The learned knowledge in the task neurons is used to initialize the created group node using a two-step process. In the first step, the weights of the group node are obtained by averaging the parameters of the concerned task nodes. This averaging is justified by the alignment constraint used in the task learning phase that ensures that the corresponding parameters learn similar concepts. Figure 1 depicts the averaging initialization using a plus symbol. In the second step, MTSL distills the information learned by multiple task nodes into the group node using an attention-based feature amalgamation method (Ye et al., 2019) referred to as **ATT**. Figure 1 depicts this amalgamation process by using arrows from task node features $F_1$ and $F_2$ to group node feature $\mathcal{F}$.

$$\mathcal{L}_{KA} = \frac{1}{N} \sum_{i}^{N} \left( F_i - ATT_i^{net}(\mathcal{F}) \odot \mathcal{F} \right)^2 \tag{2}$$

The knowledge amalgamation objective $\mathcal{L}_{KA}$ is provided in Equation 2 assuming that there are $N$ tasks grouped together. $ATT^{net}$ denotes the attention network consisting of two linear layers with an intermediate ReLU activation and a final sigmoid activation. $ATT_i^{net}(\mathcal{F})$ provides a $1 \times C$ dimensional attention vector which acts as a weight for the different channels of $\mathcal{F}$ allowing the selective distillation of task features into the group feature.

### 3.4 REMOVING TASK NEURONS

Starting from a dense set of neurons as in the initial state of MTSL provides the opportunity to leverage a rich information flow originating from diverse task information. Using neuron removal, MTSL moves towards a sparser architecture by removing task nodes that learn similar representations. These locally similar task nodes become redundant once they transfer their knowledge to the group node. The task branch is then disconnected from these redundant task nodes and connected to the group node. As defined in Section 3.1, the neurons in the task branch that now connect to the group node become task nodes. These changes are evident in the depicted next state in Figure 1.

### 3.5 MTSL ALGORITHM

Algorithm 1 presents the different phases involved in the MTSL algorithm, namely the task learning phase, the structural learning phase, and a fine-tuning phase. The task learning phase and the structural learning phase occur alternatively for $n$ number of times, followed by a final fine-tuning phase. In the task learning phase, the entire network is trained to minimize multi-task loss and maximize similarity among task nodes as described in Section 3.2. The structural learning phase involves neuron creation and neuron removal as discussed in Section 3.3 and in Section 3.4, respectively. $E_t$ determines the number of epochs for which each subsequent task learning phase is executed. Similarly, $E_s$ determines the epochs for ATT-based knowledge transfer. Considering a total training budget of $E$ epochs, the task learning phase is executed up to $E - f$ epochs where $f$ is the minimum epochs allocated for the fine-tuning phase during which the task nodes are no longer forced to align.

## 4 EXPERIMENTS

We evaluate the strengths of our approach using two datasets, namely **Cityscapes** (Cordts et al., 2016) and **NYUv2** (Silberman et al., 2012). The Cityscapes dataset is an outdoor driving scenes dataset consisting of 2975 training images and 500 validation images. The images are reshaped to a resolution of $256 \times 512$ in both training and validation. The NYUv2 dataset consists of indoor scenes with a total of 795 training and 654 validation images, respectively. Cityscapes and NYUv2 datasets are also referred to as CS and NYU, respectively. We consider five dense prediction tasks, namely semantic segmentation ($\mathcal{S}$), depth estimation ($\mathcal{D}$), edge detection ($\mathcal{E}$), surface normals ($\mathcal{N}$), and autoencoder ($\mathcal{A}$). All numbers are reported on the validation set of each dataset as an average over three runs. Each of the experiments has been run on one Nvidia Tesla V100 GPU in a DGX cluster. The ResNet encoder is initialized with the ImageNet pretrained weights and the rest of the

---

**Algorithm 1:** MTSL algorithm

---

**Input:** Initial state (as depicted in Figure 1), Training budget $E$, Minimum fine-tuning budget $f$;
$n \leftarrow$ Number of structural learning phases;
Task Learning Epochs $E_t \leftarrow [t^1, t^2.., t^n], \sum_i^n E_t^i < E - f$;
Structural Learning Epochs $E_s \leftarrow [s^1, s^2.., s^n]$;
**while** $e < E - f$ **do**

    $t \leftarrow$ Next value from $E_t$;                // *Task Learning Phase*
    **for** $t$ epochs **do**
        | Train using loss $\mathcal{L}$ in Equation 1;
    **end**
    $e \leftarrow e + t$;
    Create group nodes using local task similarity;    // *Structural Learning Phase*
    Average corresponding task nodes to initialize group nodes;
    $s \leftarrow$ Next value from $E_s$;
    **for** $s$ epochs **do**
        | Use ATT to transfer knowledge in corresponding task nodes to group nodes;
    **end**
    **if** there is no more layer in all task branches **then**
        | Exit loop;
    **end**

**end**
**for** $E - e$ epochs **do**                // *Fine-Tuning Phase*
    | Fine-tune using multi-task loss $\mathcal{L}_{MTL}$;
**end**

---

weights are initialized randomly. The CKA regularization loss weight $\lambda$ is 0.2. Additional training details can be found in Section C in Appendix.

We provide multi-task performance improvement (Vandenhende et al., 2021) using Equation 3 where $m$ and $s$ represent task performance in multi-task and single task networks, respectively. When higher performance is better $l$ is 0 and when the lower performance is better $l$ is 1. We also report multi-task performance improvements in segmentation $\mathcal{S}$ and depth $\mathcal{D}$ tasks as $\Delta_{MTL}^{\mathcal{SD}}$,

$$\Delta_{MTL} = \frac{1}{T} \sum_{i=1}^{T} (-1)^{l_i} \left(M_{m,i} - M_{s,i}\right) / M_{s,i} \tag{3}$$

We compare MTSL with state-of-the-art methods (Section 4.1), evaluate the generalization, inference efficiency (Section 4.2), and robustness of MTSL (Section 4.3). We draw insights based on the converged network architecture in Section 4.4 and provide a detailed ablation study in Section 4.5.

## 4.1 COMPARISON WITH STATE-OF-THE-ART METHODS

We compare MTSL with existing multi-task architectures and methods to learn multi-task architecture branching. Implementations of Cross-stitch (Misra et al., 2016) and MTAN (Liu et al., 2019) have been taken from the repository of (Vandenhende et al., 2021) while the implementation of Learning-to-Branch (LTB; (Guo et al., 2020)) has been taken from LibMTL (Lin & Zhang, 2022). We update LTB with the resource loss proposed by Branched Multi-Task Architecture Search BMTAS (Bruggemann et al., 2020) from their open source repository to obtain the implementation of BMTAS. LTB-R and BMTAS-R refer to the results obtained by retraining the converged architecture from scratch. For all methods we use the same hyperparameter settings and use ResNet18 backbone with DeepLab head.

In Table 1, we observe that Cross-stitch obtains the best improvement (highlighted in bold) over Single Task Networks (STNs). Cross-stitch retains all separate task networks and learns task-specific adapter parameters. These adapter parameters enable the selective transfer of information from other tasks. Additionally, since there is no explicit parameter sharing, task interference can be mitigated,

Table 1: IID Generalization and inference efficiency comparisons between MTSL and state-of-the-art methods, namely Cross-stitch, MTAN, LTB and BMTAS. LTB-R and BMTAS-R represent results obtained by retraining the final converged architecture of LTB and BMTAS, respectively. # (M) denotes parameter count in millions and GMac denotes Giga multiply-accumulate.

| Network | CS | | | | NYU | | | |
|---|---|---|---|---|---|---|---|---|
| | $\Delta_{MTL}^{\mathcal{SD}} \uparrow$ | $\Delta_{MTL} \uparrow$ | # (M) | GMac | $\Delta_{MTL}^{\mathcal{SD}} \uparrow$ | $\Delta_{MTL} \uparrow$ | # (M) | GMac |
| STN | - | - | 79.50 | 26.73 | - | - | 79.51 | 29.24 |
| One-Net | $-0.84_{\pm 0.15}$ | $-1.22_{\pm 0.04}$ | 34.79 | 7.70 | $-0.79_{\pm 0.11}$ | $-4.57_{\pm 0.13}$ | 34.80 | 8.42 |
| Cross-stitch | $\textbf{+2.79}_{\pm 0.41}$ | $\textbf{+1.01}_{\pm 0.17}$ | 79.50 | 26.73 | $\textbf{+0.73}_{\pm 0.37}$ | $-2.18_{\pm 0.25}$ | 79.53 | 29.24 |
| MTAN | $-0.02_{\pm 0.49}$ | $-3.68_{\pm 0.64}$ | 36.61 | 11.13 | $-0.34_{\pm 0.93}$ | $-3.91_{\pm 0.54}$ | 36.62 | 12.18 |
| LTB-R | $+0.49_{\pm 0.40}$ | $+0.11_{\pm 0.17}$ | 68.15 | 19.81 | $-0.13_{\pm 0.10}$ | $-0.23_{\pm 0.08}$ | 79.17 | 25.20 |
| BMTAS-R | $+0.25_{\pm 0.57}$ | $-0.02_{\pm 0.26}$ | 68.30 | 21.03 | $-0.10_{\pm 0.29}$ | $\textbf{-0.21}_{\pm 0.28}$ | 79.17 | 25.20 |
| MTSL-R | $+0.17_{\pm 0.47}$ | $-0.22_{\pm 0.20}$ | 68.37 | 12.00 | $-0.39_{\pm 0.46}$ | $-3.54_{\pm 0.23}$ | 59.98 | 11.95 |
| LTB | $-2.95_{\pm 0.42}$ | $-1.79_{\pm 0.18}$ | 68.15 | 19.81 | $-4.78_{\pm 0.40}$ | $-5.35_{\pm 0.31}$ | 79.17 | 25.20 |
| BMTAS | $-3.84_{\pm 0.39}$ | $-2.19_{\pm 0.11}$ | 68.30 | 21.03 | $-4.00_{\pm 0.74}$ | $-5.07_{\pm 0.42}$ | 79.17 | 25.20 |
| MTSL | $\underline{-0.23}_{\pm 0.85}$ | $\underline{-0.34}_{\pm 0.37}$ | 68.37 | 12.00 | $\underline{-0.06}_{\pm 0.34}$ | $\underline{-3.21}_{\pm 0.21}$ | 59.98 | 11.95 |

leading to increased performance. However, Cross-stitch loses the inference time and memory advantage of using shared parameters. When comparing MTSL with methods that learn branching structures (last group), MTSL outperforms both LTB and BMTAS by a considerable margin in both datasets. For instance, in $\Delta_{MTL}^{\mathcal{SD}}$, MTSL outperforms LTB by 2.72% and 4.72% and BMTAS by 3.61% and 3.94% in Cityscapes and NYUv2 datasets. Also, with the aid of CKA regularization, MTSL leads to a more inference efficient architecture (visualized in Section E) as evidenced from GMac. LTB-R, BMTAS-R and MTSL-R are versions of LTB, BMTAS and MTSL respectively where the final converged architecture is reinitialized and retrained. With additional training, LTB-R and BMTAS-R can obtain a marginal gain over MTSL. However, they deviate from structural learning as the original trained weights are no longer relevant and are discarded. MTSL-R is more efficient in terms of inference and shows a performance comparable to that of LTB-R and BMTAS-R. Note that MTSL is designed for structural learning and shows clear performance and inference efficiency improvements over LTB and BMTAS.

The methods discussed so far only consider learning branching structure in the encoder. For the upcoming experiments, we extend MTSL to an encoder-decoder architecture with a total of ten locations, six in the encoder and four in the decoder where task nodes can participate in neuron creation and removal. The encoder is based on ResNet18 and the decoder has ResNet blocks.

## 4.2 IID GENERALIZATION AND INFERENCE EFFICIENCY

MTSL algorithm begins from **STN**s and could potentially end in a network with all encoder and decoder layers shared between all tasks (referred to as **One-Net**). These two possible networks are considered as the baselines for evaluating our approach. Table 2 shows the generalization performance of the baselines and MTSL. MTSL achieves better generalization than One-Net in most cases, while being close to One-Net in inference efficiency, as seen in the GMac and parameter count columns. In NYUv2, MTSL even rivals the performance of STN in both $\mathcal{S}$ and $\mathcal{D}$ tasks. We hypothesize that the improved generalization can be attributed to brain-inspired aspects of the MTSL algorithm, such as local task similarity and the change of dense to sparse architecture.

The training time of MTSL is higher than that of One-Net because of the training epochs required for knowledge amalgamation. However, this only adds 34 additional epochs in training where only the network parts up until the task nodes is involved in the computation. For a fair comparison, we train One-Net for 34 more epochs and get the One-Net-L baseline. We note that MTSL also improves generalization over One-Net-L. Overall, we see that with only a fractional increase in training costs, MTSL algorithm provides a learned network with better generalization.

Table 2: IID generalization and inference efficiency comparisons between MTSL and baselines. MTSL performs better than One-Net in most cases and achieves an inference efficiency close to One-Net. # (M) denotes the number of parameters in millions.

| | Network | $\mathcal{S}\uparrow$ | $\mathcal{D}\downarrow$ | $\mathcal{E}\downarrow$ | $\mathcal{N}\uparrow$ | $\mathcal{A}\downarrow$ | $\Delta^{\mathcal{SD}}_{MTL}\uparrow$ | $\Delta_{MTL}\uparrow$ | # (M) | GMac |
|---|---|---|---|---|---|---|---|---|---|---|
| CS | STN | $60.87_{\pm0.78}$ | $6.37_{\pm0.02}$ | $0.03_{\pm0.00}$ | $0.61_{\pm0.00}$ | $0.05_{\pm0.00}$ | - | - | 107.10 | 31.03 |
| | One-Net | $60.34_{\pm0.37}$ | $6.76_{\pm0.04}$ | $\mathbf{0.04}_{\pm0.00}$ | $0.59_{\pm0.00}$ | $\mathbf{0.06}_{\pm0.00}$ | $-3.47_{\pm0.80}$ | $-9.65_{\pm0.46}$ | 21.70 | 7.06 |
| | One-Net-L | $60.71_{\pm0.21}$ | $6.75_{\pm0.03}$ | $\mathbf{0.04}_{\pm0.00}$ | $0.59_{\pm0.00}$ | $\mathbf{0.06}_{\pm0.00}$ | $-3.15_{\pm0.62}$ | $-9.39_{\pm0.57}$ | 21.70 | 7.06 |
| | MTSL | $\mathbf{60.68}_{\pm0.10}$ | $\mathbf{6.52}_{\pm0.03}$ | $\mathbf{0.04}_{\pm0.00}$ | $\mathbf{0.60}_{\pm0.00}$ | $\mathbf{0.06}_{\pm0.00}$ | $\mathbf{-1.35}_{\pm0.70}$ | $\mathbf{-7.04}_{\pm0.56}$ | 22.86 | 8.96 |
| NYU | STN | $35.63_{\pm0.53}$ | $52.70_{\pm0.25}$ | $0.06_{\pm0.00}$ | $0.80_{\pm0.00}$ | $0.14_{\pm0.00}$ | - | - | 107.10 | 33.99 |
| | One-Net | $34.15_{\pm0.15}$ | $53.44_{\pm0.39}$ | $\mathbf{0.06}_{\pm0.00}$ | $\mathbf{0.74}_{\pm0.00}$ | $0.17_{\pm0.01}$ | $-2.77_{\pm0.88}$ | $-9.82_{\pm1.87}$ | 21.70 | 7.77 |
| | One-Net-L | $34.42_{\pm0.17}$ | $53.36_{\pm0.31}$ | $\mathbf{0.06}_{\pm0.00}$ | $\mathbf{0.74}_{\pm0.00}$ | $0.17_{\pm0.01}$ | $-2.77_{\pm0.70}$ | $-9.82_{\pm1.67}$ | 21.70 | 7.77 |
| | MTSL | $\mathbf{35.50}_{\pm0.25}$ | $\mathbf{52.46}_{\pm0.23}$ | $\mathbf{0.06}_{\pm0.00}$ | $0.73_{\pm0.00}$ | $\mathbf{0.16}_{\pm0.00}$ | $\mathbf{+0.06}_{\pm1.03}$ | $\mathbf{-6.21}_{\pm0.90}$ | 22.81 | 9.89 |

Table 3: Robustness to natural corruptions under four categories. MTSL shows better robustness compared to One-Net in most cases.

| | Network | $\Delta^{\mathcal{SD}}_{MTL}\uparrow$ | | | | $\Delta_{MTL}\uparrow$ | | | |
|---|---|---|---|---|---|---|---|---|---|
| | | Noise | Blur | Weather | Digital | Noise | Blur | Weather | Digital |
| CS | One-Net | -5.78 | -1.32 | -3.81 | -2.15 | **+3.02** | -7.73 | -1.36 | -12.22 |
| | MTSL | **-3.03** | **+0.21** | **+0.49** | **+1.42** | +2.92 | **-3.72** | **+2.60** | **-7.07** |
| NYU | One-Net | **+10.25** | -3.71 | -3.27 | -2.72 | **+15.72** | -8.51 | -11.79 | -10.35 |
| | MTSL | +1.92 | **-2.65** | **-0.84** | **-1.14** | +13.50 | **-8.03** | **+3.22** | **-7.54** |

## 4.3 ROBUSTNESS TO NATURAL CORRUPTIONS

MTSL provides an improved generalization, likely due to its motivation originating from abstract notions of the brain. Given that the brain is effective in discerning noise from semantics, a natural question to ask is whether MTSL can lead to improvements in robustness to natural corruptions. Table 3 shows the robustness of the baselines and MTSL to various natural corruptions (Hendrycks & Dietterich, 2019) categorized into four types, namely noise, blur, weather and digital. Under each corruption category, the average across five different severity levels is taken. We see that the MTSL network exhibits better robustness compared to One-Net in most cases, especially in the weather and digital category of the Cityscapes dataset. These results further demonstrate that MTSL presents a compelling case for the utility of drawing inspirations from the brain.

## 4.4 CONVERGED NETWORK ARCHITECTURE AND TASK GROUPS

Assumptions about the data used for training play a pivotal role in determining the learned representations. In the brain, exposure to the nature of experiences determines the way neural circuitry develops (Kudithipudi et al., 2022). Thus, the resultant architecture obtained with MTSL on two datasets would likely differ. Figure 2 visualizes the One-Net architecture and the learned architectures in Cityscapes and NYUv2. Evidently, after the first branch from layer six, the task groups and the branching structure emerging on the two datasets are different. Semantic segmentation ($\mathcal{S}$) and edge detection ($\mathcal{E}$) grouped together in both datasets follows intuition as edge detection requires predictions of semantic edges. However, the emergence of certain groups of tasks such as depth and autoencoder in NYUv2 is counter-intuitive. In addition to task relationships, these results show that local task similarity and inherent biases in the dataset can also impact the optimal architecture. In Section D, we analyze the sensitivity of the converged architecture to random seeds and initialization.

## 4.5 ABLATION STUDY

To determine the effectiveness of the different components involved in MTSL, we perform systematic evaluations and provide the results in Table 4. When alignment is not used, the task representations diverge largely around layers 3 and 4. As a result, the task nodes no longer become sufficiently similar to be fused. Therefore, the resultant architectures in the first three rows perform similarly

Table 4: Effect of alignment (Align), average initialization (Avg), and attention-based knowledge amalgamation (ATT). In the first three rows, without alignment, the resultant networks remain close to the STN. In the last three rows, the alignment results in networks closer to One-Net.

| Align | Avg | ATT | CS | | | | NYU | | | |
|---|---|---|---|---|---|---|---|---|---|---|
| | | | $\Delta_{MTL}^{SD} \uparrow$ | $\Delta_{MTL} \uparrow$ | # (M) | GMac | $\Delta_{MTL}^{SD} \uparrow$ | $\Delta_{MTL} \uparrow$ | # (M) | GMac |
| | ✓ | | -0.35 | -2.25 | 97.37 | 17.02 | +0.38 | -4.24 | 69.94 | 17.87 |
| | | ✓ | -0.05 | -1.92 | 97.37 | 17.02 | +0.57 | -4.23 | 95.97 | 17.87 |
| | ✓ | ✓ | -0.39 | -2.12 | 95.97 | 16.30 | +0.31 | -4.25 | 95.97 | 17.87 |
| ✓ | ✓ | | -2.29 | -7.67 | 22.56 | 8.71 | -1.26 | -6.68 | 22.97 | 10.32 |
| ✓ | | ✓ | -3.62 | -9.66 | **22.51** | **8.60** | -2.36 | -6.71 | 23.10 | 10.90 |
| ✓ | ✓ | ✓ | **-1.35** | **-7.04** | 22.86 | 8.96 | **+0.06** | **-6.21** | **22.81** | **9.89** |

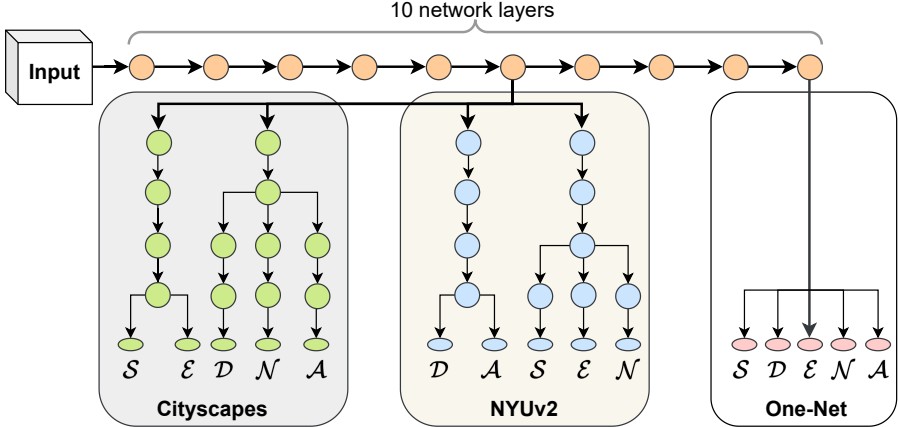

Figure 2: Visualization of the One-Net architecture and the MTSL resultant architectures. In both datasets, tasks branch out at layer six but differences emerge in the branching structure in the following layers. The oval shapes at the end of each path represents the task heads.

and have inference requirements close to STN. On the other hand, in the last three rows, with the help of alignment, the resultant architectures undergo neuron creation and removal in more layers and approach close to the inference efficiency of One-Net. Of these, the last row constitutes all the components used in MTSL and results in the best performance. Note that although we use CKA for alignment and attention for knowledge amalgamation, various other methods can also be used.

## 5 CONCLUSION

Inspired by the notion that both the structure of the neural circuits and the associated synaptic strengths change together in the brain, we proposed *Multi-Task Structural Learning (MTSL)*. MTSL relies on local similarity-induced creation of group neurons and removal of task neurons. We showed that MTSL results in networks with improved generalization and robustness while improving the inference efficiency. We discussed the dependence of converged network architectures on local task similarity and dataset. We studied the role of the different components in MTSL and found that enforcing local task similarities results in architectures with better inference efficiency.

**Limitations.** A fixed epoch schedule is used to transition between different learning phases of MTSL. Instead, it can be explored to automatically determine the periods in training in which structural learning is required. In this regard, the local inter-neuron activity in the brain could provide useful cues to automatically drive structural changes. Further, local similarity should likely be an emergent result of inter-neuron activity contrary to being explicitly enforced as in MTSL. MTSL does not use synaptogenesis and synaptic pruning, thereby limiting its ability to learn connections. Despite these limitations, MTSL serves as an indication of the potential of extending simplified notions from neuroscience into multi-task learning.

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

## A    ADDITIONAL RELATED WORKS

MTSL relies on local task similarity to drive group neuron creation and removal of the corresponding task neurons. The learned convolutional filters in different task branches might not align one-on-one due to the permutation invariance of convolutional neural networks (Wang et al., 2020a). Existing works (Wang et al., 2020a; Leontev et al., 2019; Singh & Jaggi, 2020; He et al., 2018) use different ways to align the corresponding layers of two models to counteract the permutation invariance. MTSL uses CKA (Kornblith et al., 2019) to align neurons based on representation similarity. Knowledge amalgamation approaches (Li & Bilen, 2020; Shen et al., 2019; Luo et al., 2020; 2019; He et al., 2018) address distilling the knowledge from multiple learned teachers into a single student. (Ye et al., 2019) create task-specific coding at a layer in the student network using a small network for feature distillation. MTSL uses this feature distillation process to exploit the knowledge of the task neurons set to be removed.

## B    GROUPING ALGORITHM

At the beginning of the structural learning phase, the task nodes are grouped according to the local similarity between task node features. The grouping algorithm used to provide this grouping decision is detailed in Algorithm 2. This algorithm provides the best possible task grouping, which is subsequently used to determine the creation of group nodes and the removal of task nodes.

---

**Algorithm 2:** Grouping

**Input:** Task set $\mathcal{T}$ and Similarity $S_{ij}, i, j \in \mathcal{T}, i \neq j$;
Group $G \leftarrow \{\text{tasks}\}$;
Grouping $\mathcal{G} \leftarrow \{\text{groups}\}$;
Task value in group $G$ is $\mathcal{V}_t \leftarrow \frac{1}{i} \sum_i S_{ti}$, where $i \subset G \backslash t$;
Group value $\mathcal{V}_G \leftarrow \frac{1}{t} \sum_t \mathcal{V}_t$, where t is the #tasks in group ;
Unique Grouping $\mathbb{G} \subset \mathcal{G}$, such that all tasks are present exactly once;
**for** *all unique groupings* **do**
$\quad \mid \quad \mathcal{V}_{\mathbb{G}} \leftarrow \frac{1}{g} \sum_g \mathcal{V}_g$, where g is the #groups in grouping;
**end**
Final grouping $\leftarrow$ grouping with maximum $\mathcal{V}_{\mathbb{G}}$;
**Result:** Final grouping

---

## C    ADDITIONAL TRAINING DETAILS

We provide additional training details to aid with reproducibility. The five tasks used in our experiments $\mathcal{S}, \mathcal{D}, \mathcal{E}, \mathcal{N}$ and $\mathcal{A}$ are evaluated with mIoU, RMSE, BCE (Binary Cross Entropy Error), Cosine Similarity, and MSE, respectively. The edge detection task $\mathcal{E}$ concerns the predictions of semantic edges in the scene. We use an encoder-decoder architecture where the first four layers of the encoder are initialized with ImageNet pretrained weights, while the last layer of the encoder and the decoder are initialized randomly.

For both baselines and MTSL, we use the same training hyperparameters and evaluation metrics. For training, we use the Adam optimizer with a learning rate of 1e-4 for 80 epochs. We use the step-wise learning schedule with steps at epochs 60 and 70. The batch size used is 16, a weight decay of 5e-5 and we equally weigh all task losses (all task losses have a weight of 1). In addition to averaging all parameters to initialize the group node, we also average the optimizer state of the task nodes to get the optimizer state of the group node. All other parameters that are not removed retain their weights, as well as their optimizer state. The grouping threshold $\gamma$ used is 0.8 and the number of structural learning phases is set to 10. The task learning epochs for subsequent phases are 2, 2, 2, 2, 4, 4, 8, 8, 8, and 8 and knowledge amalgamation is done for 1, 1, 2, 2, 2, 2, 4, 4, 8, and 8 epochs in the subsequent structural learning phases. The task learning epochs and knowledge amalgamation epochs are increased progressively based on nature of task representations learned at different stages of training. In early training, all task nodes are in early layers and need to learn low-level features. Our intuition here is that these generic features are likely common across tasks and can be learned fast.

The former suggests our use of low epochs in the structural learning phase, and the latter suggests the task learning phase early in the training. As we progress through the training, more epochs would be required to learn task-specific features, suggesting the increase in task-specific phase epochs. Also, later in the training, features of each task likely diverge becoming more task-specific, suggesting the increase in structural learning phase epochs.

## D  SENSITIVITY ANALYSIS

### D.1  SENSITIVITY OF THE CONVERGED ARCHITECTURES

In the main paper, we showed that the converged architecture differs between the two datasets Cityscapes and NYUv2. This suggests that the algorithm is sensitive to the dataset. In addition, we observe that the converged architectures are also sensitive to the initialization of the encoder and to random seeds. We tabulate the different architectures obtained in Table 5. The layer column lists the layers in the network, and the corresponding entries in the remaining columns show which tasks have been grouped. For instance, $[\mathcal{D}, \mathcal{N}, \mathcal{A}]$ means $\mathcal{D}, \mathcal{N}$, and $\mathcal{A}$ are grouped together.

Table 5: Sensitivity of the converged architecture to different initialization of the encoder and random seeds in both datasets. The random seeds groupings using ImageNet pretrained weights for the encoder.

|  | Layer | Seed 0 | Seed 1 | Seed 2 | Random Seed 2 |
|---|---|---|---|---|---|
| CS | 1-6 | $[\mathcal{S}, \mathcal{E}, \mathcal{D}, \mathcal{N}, \mathcal{A}]$ | $[\mathcal{S}, \mathcal{E}, \mathcal{D}, \mathcal{N}, \mathcal{A}]$ | $[\mathcal{S}, \mathcal{E}, \mathcal{D}, \mathcal{N}, \mathcal{A}]$ | $[\mathcal{S}, \mathcal{E}, \mathcal{D}, \mathcal{N}, \mathcal{A}]$ |
| | 7,8 | $[\mathcal{S}, \mathcal{E}, \mathcal{A}], [\mathcal{D}, \mathcal{N}]$ | $[\mathcal{S}, \mathcal{E}], [\mathcal{D}, \mathcal{N}, \mathcal{A}]$ | $[\mathcal{S}, \mathcal{E}], [\mathcal{D}, \mathcal{N}, \mathcal{A}]$ | $[\mathcal{S}, \mathcal{E}, \mathcal{D}, \mathcal{N}, \mathcal{A}]$ |
| | 9 | $[\mathcal{S}, \mathcal{E}, \mathcal{A}], [\mathcal{D}], [\mathcal{N}]$ | $[\mathcal{S}, \mathcal{E}], [\mathcal{D}], [\mathcal{N}], [\mathcal{A}]$ | $[\mathcal{S}, \mathcal{E}], [\mathcal{D}, \mathcal{N}, \mathcal{A}]$ | $[\mathcal{S}, \mathcal{E}, \mathcal{D}, \mathcal{N}, \mathcal{A}]$ |
| | 10 | $[\mathcal{S}, \mathcal{E}, \mathcal{A}], [\mathcal{D}], [\mathcal{N}]$ | $[\mathcal{S}, \mathcal{E}], [\mathcal{D}], [\mathcal{N}], [\mathcal{A}]$ | $[\mathcal{S}, \mathcal{E}], [\mathcal{D}, \mathcal{N}], [\mathcal{A}]$ | $[\mathcal{S}, \mathcal{E}, \mathcal{D}, \mathcal{N}, \mathcal{A}]$ |
| NYU | 1-6 | $[\mathcal{S}, \mathcal{E}, \mathcal{D}, \mathcal{N}, \mathcal{A}]$ | $[\mathcal{S}, \mathcal{E}, \mathcal{D}, \mathcal{N}, \mathcal{A}]$ | $[\mathcal{S}, \mathcal{E}, \mathcal{D}, \mathcal{N}, \mathcal{A}]$ | $[\mathcal{S}, \mathcal{E}, \mathcal{D}, \mathcal{N}, \mathcal{A}]$ |
| | 7,8,9 | $[\mathcal{S}, \mathcal{E}], [\mathcal{D}, \mathcal{N}, \mathcal{A}]$ | $[\mathcal{S}, \mathcal{E}, \mathcal{N}], [\mathcal{D}, \mathcal{A}]$ | $[\mathcal{S}, \mathcal{E}], [\mathcal{D}, \mathcal{N}, \mathcal{A}]$ | $[\mathcal{S}, \mathcal{E}, \mathcal{D}, \mathcal{N}, \mathcal{A}]$ |
| | 10 | $[\mathcal{S}, \mathcal{E}], [\mathcal{D}], [\mathcal{N}], [\mathcal{A}]$ | $[\mathcal{S}], [\mathcal{E}], [\mathcal{N}], [\mathcal{D}, \mathcal{A}]$ | $[\mathcal{S}], [\mathcal{E}], [\mathcal{D}, \mathcal{N}, \mathcal{A}]$ | $[\mathcal{S}, \mathcal{E}, \mathcal{D}, \mathcal{N}, \mathcal{A}]$ |

### D.2  SENSITIVITY TO GROUPING THRESHOLD

The grouping threshold determines whether or not a group of task nodes are similar enough to lead to the creation of a group node and eventual removal of the concerned task nodes. The sensitivity of MTSL to different grouping thresholds is shown in Table 6. A high grouping threshold would mean that task nodes are never grouped, and the resultant architecture will be similar to STN in terms of GMac and parameters. However, the performance would still be different due to the use of CKA to regularize the task representations. A low grouping threshold would mean that task nodes are always grouped leading to architectures more similar to One-Net. We observe that for a threshold of 0.7 or less, MTSL leads to One-Net architecture, but the performance is not sufficient. A threshold of 0.9 or higher leads to slow inference networks. A threshold of 0.8 provides the right trade-off and is used for all experiments in the main paper.

### D.3  SENSITIVITY TO NUMBER OF TASKS

We evaluate whether or not MTSL leads to improvements over One-Net when only 4 out of the 5 tasks are used. To this end, we train multi-task networks for 4 tasks, namely $\mathcal{S}, \mathcal{E}, \mathcal{D}$ and $\mathcal{N}$. The results are tabulated in 7. We observe that even when there are 4 tasks, MTSL improves over One-Net.

## E  CONVERGED ARCHITECTURES

In Table 1, we provide results for state-of-the-art LTB and BMTAS methods. Here, we visualize the converged architectures of LTB and BMTAS along with those of MTSL and provide inferences. Figure 3 illustrates the converged architectures on both the Cityscapes and NYUv2 dataset. We see that LTB and BMTAS generally converged to a larger architecture in comparison to MTSL. MTSL is

Table 6: Sensitivity of MTSL to the grouping threshold. For example, MTSL-0.1 denotes MTSL with a grouping threshold of 0.1. # (M) denotes the number of parameters in millions.

| | Network | $\mathcal{S}\uparrow$ | $\mathcal{D}\downarrow$ | $\mathcal{E}\downarrow$ | $\mathcal{N}\uparrow$ | $\mathcal{A}\downarrow$ | $\Delta_{MTL}^{\mathcal{SD}}\uparrow$ | $\Delta_{MTL}\uparrow$ | # (M) | GMac |
|---|---|---|---|---|---|---|---|---|---|---|
| | STN | 61.95 | 6.38 | 0.0341 | 0.6108 | 0.0535 | - | - | 107.10 | 31.03 |
| CS | One-Net | 60.02 | 6.71 | 0.0421 | 0.5941 | 0.0620 | -4.10 | -10.06 | 21.70 | 7.06 |
| | MTSL-0.1 | 59.98 | 6.75 | 0.0439 | 0.5955 | 0.0626 | -4.48 | -11.44 | 21.70 | 7.06 |
| | MTSL-0.2 | 59.97 | 6.68 | 0.0433 | 0.5956 | 0.0629 | -3.92 | -10.98 | 21.70 | 7.06 |
| | MTSL-0.3 | 59.97 | 6.70 | 0.0437 | 0.5961 | 0.0629 | -4.06 | -11.25 | 21.70 | 7.06 |
| | MTSL-0.4 | 60.32 | 6.67 | 0.0436 | 0.5960 | 0.0626 | -3.59 | -10.90 | 21.70 | 7.06 |
| | MTSL-0.5 | 59.95 | 6.68 | 0.0437 | 0.5963 | 0.0626 | -3.95 | -11.09 | 21.70 | 7.06 |
| | MTSL-0.6 | 60.31 | 6.72 | 0.0431 | 0.5967 | 0.0628 | -3.98 | -10.81 | 21.70 | 7.06 |
| | MTSL-0.7 | 59.48 | 6.68 | 0.0432 | 0.5965 | 0.0625 | -4.32 | -10.90 | 21.70 | 7.06 |
| | MTSL-0.8 | 60.64 | 6.52 | 0.0415 | 0.6023 | 0.0565 | -2.13 | -6.59 | 22.99 | 9.53 |
| | MTSL-0.9 | 60.04 | 6.46 | 0.0359 | 0.6074 | 0.0559 | -2.12 | -2.91 | 95.97 | 16.30 |
| | MTSL-1.0 | 60.73 | 6.46 | 0.0340 | 0.6105 | 0.0538 | -1.61 | -0.71 | 107.10 | 31.03 |
| | STN | 35.36 | 52.36 | 0.0566 | 0.7966 | 0.1401 | - | - | 107.10 | 33.99 |
| NYU | One-Net | 33.99 | 53.26 | 0.0634 | 0.7378 | 0.1557 | -2.80 | -7.22 | 21.70 | 7.77 |
| | MTSL-0.1 | 34.48 | 53.29 | 0.0634 | 0.7330 | 0.1859 | -2.13 | -11.39 | 21.70 | 7.77 |
| | MTSL-0.2 | 33.99 | 52.95 | 0.0635 | 0.7328 | 0.1840 | -2.49 | -11.30 | 21.70 | 7.77 |
| | MTSL-0.3 | 34.69 | 52.92 | 0.0638 | 0.7337 | 0.1824 | -1.48 | -10.75 | 21.70 | 7.77 |
| | MTSL-0.4 | 34.07 | 53.23 | 0.0634 | 0.7346 | 0.1772 | -2.65 | -10.32 | 21.70 | 7.77 |
| | MTSL-0.5 | 34.19 | 52.94 | 0.0633 | 0.7349 | 0.1861 | -2.20 | -11.36 | 21.70 | 7.77 |
| | MTSL-0.6 | 34.27 | 53.07 | 0.0636 | 0.7360 | 0.1860 | -2.21 | -11.43 | 21.70 | 7.77 |
| | MTSL-0.7 | 34.53 | 53.22 | 0.0638 | 0.7330 | 0.1888 | -2.00 | -11.89 | 21.70 | 7.77 |
| | MTSL-0.8 | 34.63 | 52.81 | 0.0586 | 0.7309 | 0.1649 | -1.46 | -6.48 | 22.84 | 10.11 |
| | MTSL-0.9 | 35.22 | 52.25 | 0.0587 | 0.7149 | 0.1504 | -0.10 | -4.30 | 95.97 | 17.87 |
| | MTSL-1.0 | 35.71 | 52.88 | 0.0565 | 0.7921 | 0.1344 | 0.00 | 0.74 | 107.10 | 33.99 |

Table 7: Sensitivity of MTSL to the number of tasks. # (M) denotes the number of parameters in millions.

| | Network | $\mathcal{S}\uparrow$ | $\mathcal{D}\downarrow$ | $\mathcal{E}\downarrow$ | $\mathcal{N}\uparrow$ | $\Delta_{MTL}^{\mathcal{SD}}\uparrow$ | $\Delta_{MTL}\uparrow$ | # (M) | GMac |
|---|---|---|---|---|---|---|---|---|---|
| | STN | 59.84 | 6.43 | 0.0341 | 0.6114 | - | - | 85.68 | 24.80 |
| CS | One-Net | 60.38 | 6.79 | 0.0415 | 0.5966 | -2.31 | -7.18 | 21.63 | 6.83 |
| | MTSL | **60.41** | **6.43** | **0.0360** | **0.6068** | **0.45** | **-1.35** | 77.33 | 13.75 |
| | STN | 35.41 | 53.10 | 0.0572 | 0.7881 | - | - | 85.68 | 27.18 |
| NYU | One-Net | 34.37 | 53.29 | 0.0623 | **0.7427** | -1.65 | -4.49 | 21.63 | 7.51 |
| | MTSL | **35.95** | **51.69** | **0.0585** | 0.7127 | **2.08** | **-1.92** | 77.33 | 15.09 |

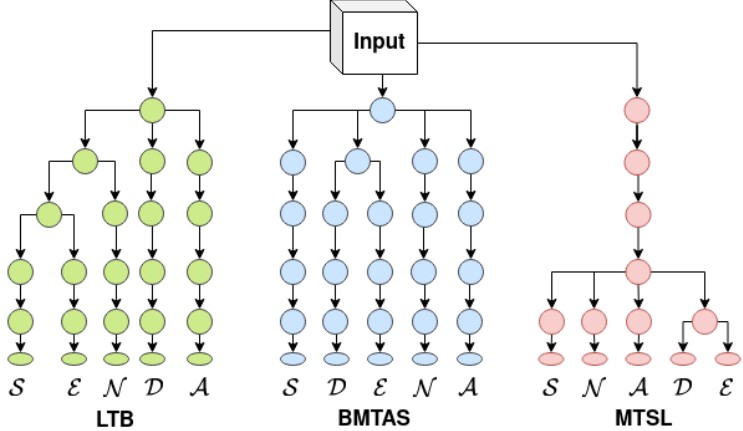

(a) Converged architecture on Cityscapes dataset.

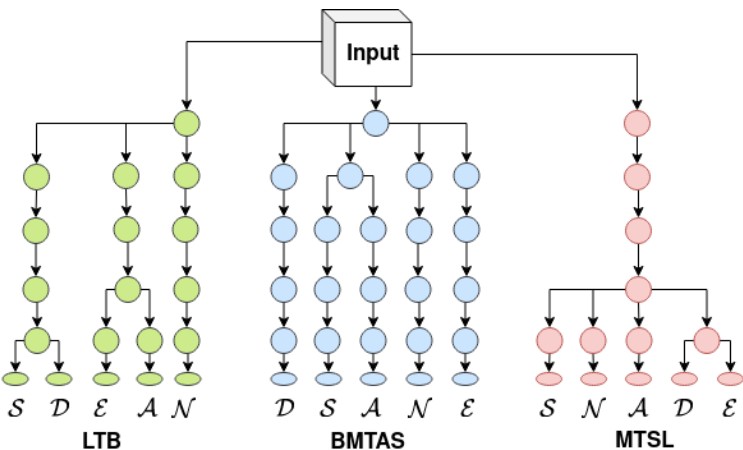

(b) Converged architecture on NYUv2 dataset.

Figure 3: Illustration of the converged architecture of LTB, BMTAS, and MTSL. In each figure, each row of circles represents a layer in the encoder. At the last layer of the encoder, each task branch out with their own heads.

able to learn a more efficient architecture, likely because of the use of the CKA regularization term, which forces tasks to learn similar representations, leading to more sharing across tasks.

## F  CENTERED KERNEL ALIGNMENT (CKA)

Centered Kernel Alignment (CKA) provides the similarity between two layers by computing the similarity between the output representations of the said layers. Let X and Y represent the output representations obtained for N images by the two layers to be compared. X and Y are then transformed by taking the mean across the spatial dimension to obtain N×C dimensional representations. We then use the unbiased estimate (Nguyen et al., 2021) to obtain the CKA. First, the gram matrices of the two representations $G_X = XX^T$ and $G_Y = YY^T$ are calculated and centered. CKA is then obtained using Equation 4.

$$CKA = \frac{G_X.G_Y}{||G_X||_F \times ||G_Y||_F} \tag{4}$$

In MTSL, to enforce the similarity between tasks, the CKA regularization term is calculated between pairs of task representations by using the current training minibatch. CKA similarities for grouping

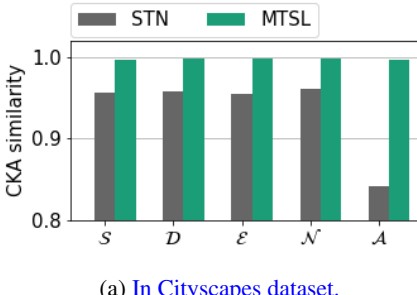

(a) In Cityscapes dataset.

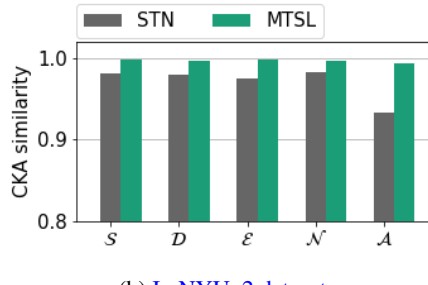

(b) In NYUv2 dataset.

Figure 4: Task vs the total similarity of the corresponding task representations with all other task representations at the output of the first layer. MTSL leads to higher CKA similarities with the aid of CKA regularization, as shown by the taller green bars.

Table 8: Sensitivity of BMTAS to the resource loss weight $\lambda$. For example, BMTAS-0.1 refers to $\lambda$=0.1 # (M) denotes the number of parameters in millions.

| | CS | | | | NYU | | | |
|---|---|---|---|---|---|---|---|---|
| Network | $\Delta_{MTL}^{\mathcal{SD}} \uparrow$ | $\Delta_{MTL} \uparrow$ | # (M) | GMac | $\Delta_{MTL}^{\mathcal{SD}} \uparrow$ | $\Delta_{MTL} \uparrow$ | # (M) | GMac |
| BMTAS-1.0 | $-3.07_{\pm 0.38}$ | $-1.89_{\pm 0.21}$ | 79.69 | 22.10 | $-3.84_{\pm 0.64}$ | $-5.06_{\pm 0.38}$ | 68.15 | 21.67 |
| BMTAS-0.1 | $-3.45_{\pm 0.19}$ | $-2.02_{\pm 0.08}$ | 79.31 | 24.25 | $-4.38_{\pm 0.98}$ | $-5.18_{\pm 0.53}$ | 79.17 | 25.20 |
| BMTAS-0.5 | $-3.19_{\pm 0.47}$ | $-1.95_{\pm 0.17}$ | 79.31 | 24.25 | $-3.83_{\pm 1.07}$ | $-5.10_{\pm 0.63}$ | 79.17 | 25.20 |
| BMTAS-0.05 | $-3.84_{\pm 0.39}$ | $-2.19_{\pm 0.11}$ | 68.30 | 21.03 | $-4.00_{\pm 0.74}$ | $-5.07_{\pm 0.42}$ | 79.17 | 25.20 |
| BMTAS-0.01 | $-3.55_{\pm 0.34}$ | $-2.10_{\pm 0.20}$ | 79.17 | 23.04 | $-3.58_{\pm 0.83}$ | $-4.90_{\pm 0.41}$ | 78.65 | 24.02 |
| BMTAS-0.02 | $-3.48_{\pm 0.18}$ | $-2.06_{\pm 0.11}$ | 78.64 | 21.96 | $-4.40_{\pm 0.87}$ | $-5.16_{\pm 0.46}$ | 76.69 | 24.17 |

decisions during training are calculated using a subset of 800 training images in Cityscapes and all 795 training images in NYUv2. To demonstrate that CKA can be optimized, we look at the sum of CKA values of a task layer with all other task layers in STN and MTSL illustrated in Figure 4. The similarities are taken using the output representations of the first layer of different single-task networks (STNs). In the case of MTSL, this similarity is taken right before tasks are grouped in the first layer. For example, the gray bar for $\mathcal{S}$ is obtained by adding the similarity of the first layer in the single task segmentation network with the first layer of all other task networks. The plot shows that MTSL is able to increase similarity between tasks, as is evident by the taller green bars, by using CKA regularization.

## G   SENSITIVITY OF BMTAS TO RESOURCE LOSS WEIGHT

We test the sensitivity of BMTAS to the resource loss weight. Both $\Delta_{MTL}^{\mathcal{SD}}$ and $\Delta_{MTL}$ values remain fairly close to each other and remain within standard deviation. Parameter count and GMac change due to the change in the effect of resource loss. We note that in the main paper, we use $\lambda$=0.05 based on the values used by the authors.

## H   ROBUSTNESS TO NATURAL CORRUPTIONS

We look at the robustness to natural corruptions of the different state-of-the-art methods and MTSL when using the ResNet encoder and DeepLab head. The results are tabulated in Table 9. Cross-stitch shows the best results among all methods likely because of the task-specific adapter blocks and task-specific networks. However, as discussed in Section 4.1, Cross-stitch has a low inference efficiency contrary to the other methods. When comparing LTB, BMTAS and MTSL, except in noise corruption, MTSL provides the most robustness. The cause of the reduced robustness of MTSL to noise corruption is currently unclear. Task interference could likely play a role, but further investigation can be done in future work.

Table 9: Robustness to natural corruptions under four categories when using the ResNet encoder and the DeepLab head. MTSL shows better robustness compared to LTB and BMTAS in most cases.

| | Network | $\Delta_{MTL}^{\mathcal{SD}} \uparrow$ | | | | $\Delta_{MTL} \uparrow$ | | | |
|---|---|---|---|---|---|---|---|---|---|
| | | Noise | Blur | Weather | Digital | Noise | Blur | Weather | Digital |
| CS | One-Net | -8.55 | -5.48 | -9.72 | -1.79 | -3.26 | -3.94 | -1.60 | -1.38 |
| | Cross-stitch | **-1.94** | **-0.17** | **-3.42** | **+2.51** | +0.14 | **-1.28** | **+3.51** | **+1.04** |
| | MTAN | -6.53 | -6.03 | -6.42 | -0.95 | -6.23 | -7.30 | -13.28 | -4.03 |
| | LTB-R | -2.52 | -2.83 | -6.08 | -2.23 | **+1.84** | -2.26 | -1.24 | -1.21 |
| | BMTAS-R | -3.48 | -2.37 | -3.63 | -1.63 | -1.80 | -1.67 | +1.01 | -0.96 |
| | MTSL-R | -5.64 | -3.35 | -7.13 | +0.95 | -2.87 | -2.97 | -1.61 | +0.03 |
| | LTB | -6.50 | -3.98 | -8.75 | -2.35 | -0.08 | -3.42 | -1.68 | -1.47 |
| | BMTAS | -7.77 | -3.49 | -11.11 | -2.63 | -1.60 | -2.81 | -5.37 | -1.55 |
| | MTSL | -9.52 | -1.68 | -4.87 | +0.56 | -3.21 | -2.46 | -1.14 | +0.16 |
| NYU | One-Net | -2.84 | -2.89 | -4.77 | -1.25 | -1.74 | -5.02 | -3.00 | -4.92 |
| | Cross-stitch | +5.82 | -1.29 | -3.31 | **+1.23** | +5.53 | -1.42 | **+1.12** | -1.48 |
| | MTAN | +2.42 | -4.87 | -3.78 | +0.36 | +0.78 | -5.57 | -4.53 | -4.26 |
| | LTB-R | +0.98 | -1.74 | -1.76 | -0.84 | +0.19 | -0.70 | -1.33 | -0.92 |
| | BMTAS-R | +2.73 | **-0.20** | **-0.44** | +0.52 | -1.16 | **-0.52** | +0.06 | **-0.14** |
| | MTSL-R | -2.13 | -3.18 | -2.94 | -0.02 | +1.56 | -3.11 | +0.20 | -3.52 |
| | LTB | +11.21 | -2.61 | -6.18 | -3.76 | **+6.49** | -4.52 | -4.07 | -5.47 |
| | BMTAS | +11.27 | -3.97 | -6.82 | -3.63 | +4.41 | -5.18 | -6.94 | -5.34 |
| | MTSL | +1.41 | -1.58 | -4.11 | +1.09 | +1.00 | -2.67 | -0.30 | -2.85 |

Table 10: Generalization and robustness comparison of MTSL against a version in which the learned architecture is reinitialized and retrained (MTSL-R).

| | Network | $\Delta_{MTL}^{\mathcal{SD}} \uparrow$ | $\Delta_{MTL} \uparrow$ | $\Delta_{MTL}^{\mathcal{SD}} \uparrow$ | | | | $\Delta_{MTL} \uparrow$ | | | |
|---|---|---|---|---|---|---|---|---|---|---|---|
| | | Generalization | | N | B | W | D | N | B | W | D |
| CS | One-Net | -3.47 | -9.65 | -5.78 | -1.32 | -3.81 | -2.15 | +3.02 | -7.73 | -1.36 | -12.22 |
| | MTSL-R | **-1.22** | **-6.71** | -3.49 | +0.15 | -4.64 | +0.51 | **+4.50** | -4.13 | +0.23 | -8.33 |
| | MTSL | -1.35 | -7.04 | **-3.03** | **+0.21** | **+0.49** | **+1.42** | +2.92 | **-3.72** | **+2.60** | **-7.07** |
| NYU | One-Net | -2.77 | -9.82 | **+10.25** | -3.71 | -3.27 | -2.72 | +15.72 | -8.51 | -11.79 | -10.35 |
| | MTSL-R | -1.34 | -6.71 | +7.25 | **-1.84** | -2.29 | -1.42 | **+15.92** | -7.16 | +3.31 | -7.72 |
| | MTSL | **+0.06** | **-6.21** | +1.92 | -2.65 | **-0.84** | **-1.14** | +13.50 | -8.03 | +3.22 | **-7.54** |

## I   RETRAINING THE CONVERGED ARCHITECTURES

We evaluate the effectiveness of the learned architecture with and without the learned parameters. The learned architecture is reinitialized and retrained under the same training settings as MTSL. The result of this retrained architecture is presented as MTSL-R in Table 10. Both MTSL-R and MTSL provide improved results when compared with One-Net, suggesting that both the learned architecture in and of itself and the MTSL training procedure of simultaneously learning the architecture and its parameters are beneficial.

Next, we provide results for MTSL-R, where the converged architectures are retrained from scratch. Instead of retraining the architecture converged on a dataset to retrain on the same dataset, we use the converged architecture in Cityscapes to train on NYUv2 and vice versa. We refer to this switched evaluation as MTSL-R-Switch. The results are provided in Table 11. We observe that switching the converged architectures leads to marginal improvements.

Table 11: Switching of the converged architectures. # (M) denotes the number of parameters in millions.

| | Network | $\mathcal{S}\uparrow$ | $\mathcal{D}\downarrow$ | $\mathcal{E}\downarrow$ | $\mathcal{N}\uparrow$ | $\mathcal{A}\downarrow$ | $\Delta_{MTL}^{\mathcal{SD}}\uparrow$ | $\Delta_{MTL}\uparrow$ | # (M) | GMac |
|---|---|---|---|---|---|---|---|---|---|---|
| CS | STN | 61.95 | 6.38 | 0.0341 | 0.6108 | 0.0535 | - | - | 107.10 | 31.03 |
| | One-Net | 60.02 | 6.71 | 0.0421 | 0.5941 | 0.0620 | -4.10 | -10.06 | 21.70 | 7.06 |
| | MTSL-R | 60.34 | 6.46 | 0.0414 | 0.6028 | 0.0567 | -1.92 | -6.51 | 22.99 | 9.53 |
| | MTSL-R-Switch | 61.30 | 6.54 | 0.0380 | 0.5987 | 0.0569 | -1.78 | -4.67 | 22.83 | 9.20 |
| NYU | STN | 35.36 | 52.36 | 0.0566 | 0.7966 | 0.1401 | - | - | 107.10 | 33.99 |
| | One-Net | 33.99 | 53.26 | 0.0634 | 0.7378 | 0.1557 | -2.80 | -7.22 | 21.70 | 7.77 |
| | MTSL-R | 34.94 | 52.89 | 0.0585 | 0.7342 | 0.1524 | -1.10 | -4.43 | 22.84 | 10.11 |
| | MTSL-R-Switch | 35.25 | 52.66 | 0.0628 | 0.7308 | 0.1561 | -0.44 | -6.30 | 22.90 | 10.47 |

Table 12: MTSL with a different similarity metric called RSA. # (M) denotes the number of parameters in millions.

| | Network | $\mathcal{S}\uparrow$ | $\mathcal{D}\downarrow$ | $\mathcal{E}\downarrow$ | $\mathcal{N}\uparrow$ | $\mathcal{A}\downarrow$ | $\Delta_{MTL}^{\mathcal{SD}}\uparrow$ | $\Delta_{MTL}\uparrow$ | # (M) | GMac |
|---|---|---|---|---|---|---|---|---|---|---|
| CS | STN | $60.87_{\pm0.78}$ | $6.37_{\pm0.02}$ | $0.03_{\pm0.00}$ | $0.61_{\pm0.00}$ | $0.05_{\pm0.00}$ | - | - | 107.10 | 31.03 |
| | MTSL | $\mathbf{60.68}_{\pm0.10}$ | $6.52_{\pm0.03}$ | $\mathbf{0.04}_{\pm0.00}$ | $0.60_{\pm0.00}$ | $\mathbf{0.06}_{\pm0.00}$ | $-1.35_{\pm0.70}$ | $-7.04_{\pm0.56}$ | 22.86 | 8.96 |
| | MTSL-RSA | $60.61_{\pm0.39}$ | $\mathbf{6.43}_{\pm0.05}$ | $\mathbf{0.04}_{\pm0.00}$ | $0.61_{\pm0.00}$ | $\mathbf{0.06}_{\pm0.00}$ | $\mathbf{-0.72}_{\pm0.61}$ | $\mathbf{-2.36}_{\pm0.18}$ | 95.97 | 16.3 |
| NYU | STN | $35.63_{\pm0.53}$ | $52.70_{\pm0.25}$ | $0.06_{\pm0.00}$ | $0.80_{\pm0.00}$ | $0.14_{\pm0.00}$ | - | - | 107.10 | 33.99 |
| | MTSL | $\mathbf{35.50}_{\pm0.25}$ | $52.46_{\pm0.23}$ | $\mathbf{0.06}_{\pm0.00}$ | $\mathbf{0.73}_{\pm0.00}$ | $0.16_{\pm0.00}$ | $+0.06_{\pm1.03}$ | $-6.21_{\pm0.90}$ | 22.81 | 9.89 |
| | MTSL-RSA | $\mathbf{35.50}_{\pm0.20}$ | $\mathbf{52.19}_{\pm0.12}$ | $\mathbf{0.06}_{\pm0.00}$ | $0.71_{\pm0.00}$ | $\mathbf{0.15}_{\pm0.00}$ | $\mathbf{+0.31}_{\pm0.54}$ | $\mathbf{-4.77}_{\pm0.53}$ | 95.97 | 16.3 |

## J  ADDITIONAL SIMILARITY METRIC

Instead of using CKA for alignment and grouping, a variety of other similarity metrics can be used. Here, we replace CKA with Representation Similarity Analysis (RSA) and provide the results in Table 12. We observe that RSA is able to improve the results of MTSL but at the cost of additional computation.

## K  EXTENDED NUMBERS FOR COMPARISON WITH SOTA

Table 13 provides extended numbers for all the sota methods against which MTSL is compared.

## L  ROBUSTNESS TO ADVERSARIAL ATTACK

In addition to the robustness to natural corruptions discussed in the main paper, we evaluate the robustness to PGD attacks (Madry et al., 2018). We perform the attacks using four epsilon levels (0.25, 0.5, 1 and 2), step size of 1 and number of iterations determined using $\min(\epsilon + 4, \lceil 1.25\epsilon \rceil)$ (Kurakin et al., 2017). For each task $\mathcal{S}$ and $\mathcal{D}$, the corresponding task loss is used for attack. MTSL provides improved robustness over One-Net in most cases, as seen in Table 14.

## M  EXTENDED NUMBER FOR GENERALIZATION RESULTS

The generalization performance of the baselines and MTSL have been provided in Table 15 with extended digits after the decimal point where required.

## N  EXTENDED NUMBERS FOR ABLATION RESULTS

The performance of each task using different components of MTSL is provided in Table 16.

Table 13: Generalization and inference efficiency comparisons between MTSL and state-of-the-art methods, namely cross stitch (Misra et al., 2016), MTAN (Liu et al., 2019), LTB (Guo et al., 2020) and BMTAS (Bruggemann et al., 2020). LTB-R and BMTAS-R denote the results obtained by retraining the converged models of LTB and BMTAS. In MTSL-RSA, the similarity metric CKA is replaced with RSA. # (M) denotes the number of parameters in millions.

| | | Network | $\mathcal{S} \uparrow$ | $\mathcal{D} \downarrow$ | $\mathcal{E} \downarrow$ | $\mathcal{N} \uparrow$ | $\mathcal{A} \downarrow$ |
|---|---|---|---|---|---|---|---|
| | | STN | $50.77_{\pm 0.18}$ | $7.22_{\pm 0.01}$ | $0.0642$ | $0.5819$ | $0.2246$ |
| CS | | One-Net | $50.90_{\pm 0.13}$ | $7.36_{\pm 0.03}$ | $0.0650_{\pm 0.00}$ | $0.5783_{\pm 0.00}$ | $0.2302_{\pm 0.00}$ |
| | | Cross-stitch | $52.63_{\pm 0.34}$ | $7.08_{\pm 0.01}$ | $0.0644_{\pm 0.00}$ | $0.5818_{\pm 0.00}$ | $0.2250_{\pm 0.00}$ |
| | | MTAN | $51.41_{\pm 0.63}$ | $7.31_{\pm 0.01}$ | $0.0650_{\pm 0.00}$ | $0.5805_{\pm 0.00}$ | $0.2626_{\pm 0.00}$ |
| | | LTB | $49.57_{\pm 0.47}$ | $7.47_{\pm 0.02}$ | $0.0649_{\pm 0.00}$ | $0.5812_{\pm 0.00}$ | $0.2289_{\pm 0.00}$ |
| | | LTB-R | $51.29_{\pm 0.28}$ | $7.22_{\pm 0.03}$ | $0.0643_{\pm 0.00}$ | $0.5820_{\pm 0.00}$ | $0.2253_{\pm 0.00}$ |
| | | BMTAS | $48.84_{\pm 0.29}$ | $7.50_{\pm 0.03}$ | $0.0649_{\pm 0.00}$ | $0.5809_{\pm 0.00}$ | $0.2290_{\pm 0.00}$ |
| | | BMTAS-R | $51.18_{\pm 0.32}$ | $7.24_{\pm 0.06}$ | $0.0644_{\pm 0.00}$ | $0.5820_{\pm 0.00}$ | $0.2253_{\pm 0.00}$ |
| | | MTSL | $50.81_{\pm 0.59}$ | $7.26_{\pm 0.03}$ | $0.0644_{\pm 0.00}$ | $0.5820_{\pm 0.00}$ | $0.2268_{\pm 0.00}$ |
| | | MTSL-RSA | $51.10_{\pm 0.20}$ | $7.18_{\pm 0.02}$ | $0.0644_{\pm 0.00}$ | $0.5819_{\pm 0.00}$ | $0.2254_{\pm 0.00}$ |
| | | STN | $34.56_{\pm 0.19}$ | $54.17_{\pm 0.17}$ | $0.0764_{\pm 0.00}$ | $0.7712_{\pm 0.00}$ | $0.5933_{\pm 0.00}$ |
| NYU | | One-Net | $33.49_{\pm 0.38}$ | $53.35_{\pm 0.29}$ | $0.0776_{\pm 0.00}$ | $0.7041_{\pm 0.00}$ | $0.6590_{\pm 0.00}$ |
| | | Cross-stitch | $34.25_{\pm 0.23}$ | $52.90_{\pm 0.24}$ | $0.0768_{\pm 0.00}$ | $0.7010_{\pm 0.00}$ | $0.6096_{\pm 0.00}$ |
| | | MTAN | $33.79_{\pm 0.51}$ | $53.33_{\pm 0.27}$ | $0.0764_{\pm 0.00}$ | $0.7070_{\pm 0.00}$ | $0.6563_{\pm 0.00}$ |
| | | LTB | $31.92_{\pm 0.04}$ | $55.21_{\pm 0.22}$ | $0.0768_{\pm 0.00}$ | $0.6964_{\pm 0.00}$ | $0.6345_{\pm 0.00}$ |
| | | LTB-R | $34.04_{\pm 0.47}$ | $53.49_{\pm 0.40}$ | $0.0768_{\pm 0.00}$ | $0.7725_{\pm 0.00}$ | $0.5970_{\pm 0.00}$ |
| | | BMTAS | $32.09_{\pm 0.29}$ | $54.64_{\pm 0.36}$ | $0.0768_{\pm 0.00}$ | $0.6978_{\pm 0.00}$ | $0.6370_{\pm 0.00}$ |
| | | BMTAS-R | $34.30_{\pm 0.26}$ | $53.88_{\pm 0.12}$ | $0.0764_{\pm 0.00}$ | $0.7685_{\pm 0.00}$ | $0.5968_{\pm 0.00}$ |
| | | MTSL | $33.83_{\pm 0.30}$ | $53.09_{\pm 0.11}$ | $0.0776_{\pm 0.00}$ | $0.6993_{\pm 0.00}$ | $0.6233_{\pm 0.00}$ |
| | | MTSL-RSA | $33.74_{\pm 0.33}$ | $53.33_{\pm 0.31}$ | $0.0768_{\pm 0.00}$ | $0.7047_{\pm 0.00}$ | $0.6271_{\pm 0.00}$ |

Table 14: Robustness to PGD attack of the baselines and MTSL.

| | Network | $\mathcal{S}\uparrow$ | | | | $\mathcal{D}\downarrow$ | | | |
|---|---|---|---|---|---|---|---|---|---|
| | | 0.25 | 0.5 | 1 | 2 | 0.25 | 0.5 | 1 | 2 |
| CS | STN | 45.42 | 40.76 | 31.14 | 24.56 | 10.80 | 13.21 | 19.93 | 25.86 |
| | One-Net | 45.28 | 40.97 | 31.41 | 24.84 | 11.63 | 14.42 | 22.16 | 29.04 |
| | MTSL | **46.42** | **41.72** | **31.94** | **25.34** | **11.28** | **14.02** | **21.52** | **28.58** |
| NYU | STN | 21.58 | 17.96 | 11.65 | 8.24 | 85.98 | 103.13 | 150.55 | 194.52 |
| | One-Net | 20.75 | 17.31 | 11.17 | 7.83 | 85.40 | 103.67 | 149.12 | **191.13** |
| | MTSL | **21.88** | **18.25** | **11.50** | **7.90** | **85.00** | **102.85** | **148.80** | 191.21 |

Table 15: Generalization comparisons between MTSL and baselines. # (M) denotes the number of parameters in millions.

| | Network | $\mathcal{S}\uparrow$ | $\mathcal{D}\downarrow$ | $\mathcal{E}\downarrow$ | $\mathcal{N}\uparrow$ | $\mathcal{A}\downarrow$ |
|---|---|---|---|---|---|---|
| CS | STN | $60.87_{\pm0.78}$ | $6.37_{\pm0.02}$ | $0.0342_{\pm0.00}$ | $0.6106_{\pm0.00}$ | $0.0535_{\pm0.00}$ |
| | One-Net | $60.34_{\pm0.37}$ | $6.76_{\pm0.04}$ | $0.0423_{\pm0.00}$ | $0.5943_{\pm0.00}$ | $0.0616_{\pm0.00}$ |
| | One-Net-L | $60.71_{\pm0.21}$ | $6.75_{\pm0.03}$ | $0.0421_{\pm0.00}$ | $0.5942_{\pm0.00}$ | $0.0615_{\pm0.00}$ |
| | MTSL | **$60.68_{\pm0.10}$** | **$6.52_{\pm0.03}$** | **$0.0419_{\pm0.00}$** | **$0.6029_{\pm0.00}$** | **$0.0583_{\pm0.00}$** |
| NYU | STN | $35.63_{\pm0.53}$ | $52.70_{\pm0.25}$ | $0.0570_{\pm0.00}$ | $0.7955_{\pm0.00}$ | $0.1355_{\pm0.00}$ |
| | One-Net | $34.15_{\pm0.15}$ | $53.44_{\pm0.39}$ | $0.0636_{\pm0.00}$ | **$0.7370_{\pm0.00}$** | $0.1688_{\pm0.01}$ |
| | One-Net-L | $34.42_{\pm0.17}$ | $53.36_{\pm0.31}$ | $0.0634_{\pm0.00}$ | **$0.7366_{\pm0.00}$** | $0.1682_{\pm0.01}$ |
| | MTSL | **$35.50_{\pm0.25}$** | **$52.46_{\pm0.23}$** | **$0.0595_{\pm0.00}$** | $0.7298_{\pm0.00}$ | **$0.1605_{\pm0.00}$** |

Table 16: Ablation results of different tasks.

| | Align | Avg | ATT | $\mathcal{S}\uparrow$ | $\mathcal{D}\downarrow$ | $\mathcal{E}\downarrow$ | $\mathcal{N}\uparrow$ | $\mathcal{A}\downarrow$ |
|---|---|---|---|---|---|---|---|---|
| CS | | ✓ | | $60.54_{\pm1.06}$ | $6.38_{\pm0.02}$ | $0.0362_{\pm0.00}$ | $0.6070_{\pm0.00}$ | $0.0558_{\pm0.00}$ |
| | | | ✓ | $61.13_{\pm0.10}$ | $6.40_{\pm0.05}$ | $0.0359_{\pm0.00}$ | $0.6073_{\pm0.00}$ | $0.0557_{\pm0.00}$ |
| | | ✓ | ✓ | $60.95_{\pm0.34}$ | $6.43_{\pm0.02}$ | $0.0358_{\pm0.00}$ | $0.6075_{\pm0.00}$ | $0.0560_{\pm0.00}$ |
| | ✓ | ✓ | | $60.29_{\pm1.32}$ | $6.60_{\pm0.13}$ | $0.0421_{\pm0.00}$ | $0.6006_{\pm0.00}$ | $0.0584_{\pm0.00}$ |
| | ✓ | | ✓ | $59.39_{\pm1.39}$ | $6.67_{\pm0.11}$ | $0.0436_{\pm0.00}$ | $0.5974_{\pm0.00}$ | $0.0596_{\pm0.00}$ |
| | ✓ | ✓ | ✓ | $60.68_{\pm0.10}$ | $6.52_{\pm0.03}$ | $0.0419_{\pm0.00}$ | $0.6029_{\pm0.00}$ | $0.0583_{\pm0.00}$ |
| NYU | | ✓ | | $35.34_{\pm0.06}$ | $51.88_{\pm0.06}$ | $0.0589_{\pm0.00}$ | $0.7145_{\pm0.00}$ | $0.1470_{\pm0.00}$ |
| | | | ✓ | $35.54_{\pm0.44}$ | $51.97_{\pm0.47}$ | $0.0590_{\pm0.00}$ | $0.7140_{\pm0.00}$ | $0.1469_{\pm0.00}$ |
| | | ✓ | ✓ | $35.64_{\pm0.50}$ | $52.40_{\pm0.07}$ | $0.0589_{\pm0.00}$ | $0.7102_{\pm0.00}$ | $0.1461_{\pm0.00}$ |
| | ✓ | ✓ | | $34.80_{\pm0.33}$ | $52.80_{\pm0.06}$ | $0.0584_{\pm0.00}$ | $0.7331_{\pm0.00}$ | $0.1634_{\pm0.00}$ |
| | ✓ | | ✓ | $34.18_{\pm0.44}$ | $53.05_{\pm0.21}$ | $0.0588_{\pm0.00}$ | $0.7344_{\pm0.00}$ | $0.1599_{\pm0.02}$ |
| | ✓ | ✓ | ✓ | $35.50_{\pm0.25}$ | $52.46_{\pm0.23}$ | $0.0595_{\pm0.00}$ | $0.7298_{\pm0.00}$ | $0.1605_{\pm0.00}$ |

