# OpenReview forum: "Multi-Task Structural Learning using Local Task Similarity induced Neuron Creation and Removal"
_ICLR.cc/2023/Conference — Submitted to ICLR 2023_

### Official Review · Reviewer_wjTd · 2022-10-24

**Confidence:** 4
**Clarity, Quality, Novelty And Reproducibility:** The paper proposes a novel method and…
**Correctness:** 3
**Technical Novelty And Significance:** 2
**Empirical Novelty And Significance:** 2
**Recommendation:** 5

**Strength And Weaknesses:**

Strength

- It proposes a novel method for multi-task NAS while constraining the search space to the tree-structured architectures, similarly to BMTAS and LTB. Compared to BMTAS and NAS, it utilized CKA to guide task streams to be similar to each other and compute similarity between layers, which is reasonable.
- The paper provides an extensive analysis of the sensitivity of several factors, including similarity measures and hyper-parameters. It is also good that the performance under several corruptions is evaluated.

Weakness

- I am not sure about the advantage of this method over other existing methods, BMTAS and LTB.
Given the same search space of tree structures, I would like to see the accuracy comparison with BMTAS and LTB after re-initialization and re-training to evaluate the optimality of the found architecture itself. The paper provides accuracy after re-training in Table 8, but a comparison with BMTAS and LTB is unavailable.

- I am not convinced about the value of simultaneous training of the architecture and its parameters, which is claimed as one contribution of this work. Since this method requires training of single-task models for all tasks for initialization, additional one-time re-intialization and re-training of the found architecture will not be a huge burden, roughly spending a similar training cost for one single task network training. For this reason, I also think the comparison with existing methods (BMTAS, LTB) would more reasonable after the re-training.

- The method needs several hyper-parameters, including the task learning epochs, structure learning epochs. How will they affect the performance? Sensitivity analysis seems missing.

- Scalability can be another issue of this method. It needs to initialize the model with N single-task models for N tasks, needs N^2 pairwise similarity, and find optimal groupings during training.

- In Table 1, for fair comparison, BMTAS can be trained with larger lambda (resource regularizer) to match the similar compute cost.
- The details of the evaluation setting of Table 2 is unclear. Is it a cross-dataset accuracy, e.g., training on NYUv2 and testing on CityScapes?
- It would be nice to include the comparison with BMTAS and LTB (after re-training) in Table 3, to compare the robustness against the corruptions.


Related work

- [Controllable Dynamic Multi-Task Architectures, CVPR 2022] looks pretty related in the sense that it also uses task similarity to guide branching. It also searches for the tree-structured multi-task architecture but also allows dynamic control of the total compute cost.


Minor

- The purpose of explicitly evaluating the multi-task performance for semantic segmentation and depth (SD) is unclear to me.




**Summary Of The Paper:**

This paper proposes a network architecture search for a multi-task network, while specifying the search space to tree structures. It jointly trains the model structure as well as the parameters. It starts from the set of single networks and gradually groups layers across tasks if their output features are sufficiently similar. Experiments show that the proposed method outperforms the existing multi-task NAS approaches (BMTAS, LTB) without re-training.

**Summary Of The Review:**

In summary, I think this paper proposes a new method for multi-task NAS. However, I could not find an advantage over other existing multi-task NAS approaches in the method and the experiments.

---

### Official Review · Reviewer_hhES · 2022-10-27

**Confidence:** 4
**Correctness:** 2
**Technical Novelty And Significance:** 3
**Empirical Novelty And Significance:** 2
**Recommendation:** 5

**Clarity, Quality, Novelty And Reproducibility:**

The presentation of the paper needs improvement. Novelty is acceptable. No code is provided. Based on the description in the paper, reproducibility is questionable.

**Strength And Weaknesses:**

Strength
+ The development of the method tried to mimic how human brains works. It may be conceptual advantage over existing ones from the artificial intelligence perspective.
+ Large efforts made to empirical study from multiple aspect, including robustness and contribution of different components of the method.

Weakness
- The calculation of CKA only directly involves kernel, which is the pairwise similarity between examples based on activation. I do not see this helps align learned filters in two separate networks.
- The calculation of CKA is quite complicated. It may be challenging to effectively optimize the overall objective in Eq. (1).
- CKA determines the similarity among tasks. Its calculation should be detailed, for example the (number) examples involved.
- After the first round of structural learning, not all task nodes are at the same layer of the network. I do not think the way that they described for grouping is appropriate. That way likely leads to a very complicated grouping structure (not like the one that they show in Figure 2, which has nice tree structure) that is difficult to interpret, making little to no sense at all.
- It would be interesting to report the task grouping/branching learned by other methods and have a comparison with that by the proposed method.
- The empirical results (Table 1) are weak and not much supportive to the value/advantage of the proposed method.
- The manuscript is readable on overall, but with large room to improve. There are inappropriate choices of wording in many places.


**Summary Of The Paper:**

This paper presents Multi-Task Structural Learning (MTSL) framework to learn a network architecture and its parameters in a multi-task learning setting. MTSL consists of two phases, a structural learning phase and a fine-tuning phase. The former includes repetition of two alternating steps, network training followed by task grouping. Two datasets in computer vision: Cityscape and NYUv2 were used to evaluate the proposed method.

**Summary Of The Review:**

The method has conceptual merit and is innovative. However, the weaknesses in the implementation and their empirical results dampen my enthusiasm of the work.

---

### Official Review · Reviewer_2dhv · 2022-11-05

**Confidence:** 2
**Correctness:** 3
**Technical Novelty And Significance:** 2
**Empirical Novelty And Significance:** 2
**Recommendation:** 5

**Clarity, Quality, Novelty And Reproducibility:**

The work appears to be methodologically sound. Experiments are effective overall, and the method
performs favorably to the methods compared. The method combines some existing approaches in a new way, so appears moderately
novel, though I am not confident in that assessment. The writing is good overall, but there are some methodological
details that are are not clear and seem not sufficiently explained. Without available code, I am not confident a researcher
could reproduce the method. I have included some questions below that the authors might consider clarifying in the text and I hope might improve the manuscript.

From the abstract "In each of the structural learning phases, starting from the earliest layer, ...", but its not clear to
me from the text when the method "moves on" from the earliest to layer layers.

How does the method determine when to "stop" merging task nodes?  The grouping threshold gamma will clearly play a role.
I appreciate section D.2 describing the authors' experiences varying gamma.

It is unclear how or why in Figure 2, the first 6 layers are shared between the two tasks. Is it possible that MTSL could have
determined that one task shares no information / nodes with the other four tasks?  Is it always the case that early layers are
shared and later layers are split? One could argue that properties of natural images make that likely, but I wonder if it is
possible for the algorithm to produce a graph that splits and merges.

Tables [1,2,3] What is the difference between \Delta^SD_MTL and \Delta_MTL? Does SD use only the S and D tasks?

Eq 1 is not very useful, the authors might consider adding definitions for L_MTL and L_CKA for completeness.

Eq (2) What does the 'star F' notation refer to? I could not find similar notation in the associated reference Ye 2019.

"locally similar task layers" what is "locality" with respect to here?

**Strength And Weaknesses:**

The proposed work describes an effective methodology for combining N single-task networks into a single, branched network
that can perform well at multiple tasks. This enables trading-off between task performance and computational efficiency. The
experiments appear to be well done, and the authors include an informative ablation study. The results make me confident that
the method can perform well and as described.

The main weakness of the paper is the lack of clarity in parts when describing the method - I give some detailed examples in the
next section.

Some minor weaknesses:
The paper would be improved if the authors interpret the results for the reader more. For example, the text only very briefly
describes the differences between LTB-R and LTB, which is an important and interesting one. As well - why does cross-stich
perform so well, and how much has to do with the number of parameters?

I find the analogy with biological neural circuits to be unconvincing, but more concerning was the statement:
"The improvement generalization can be attributed to the brain-inspired aspects of the MTSL algorithm." I am very
skeptical of this claim, and couldn't find  evidence in the work about WHY MTSL's performance is good.
More care in separating speculation (which is fine when framed as such) from conclusions (and "can be attributed to" sounds
conclusive) would be appropriate.

**Summary Of The Paper:**

The authors present a method that "merges" N single task networks by grouping similar neurons. The training process
alternates between "task learning" during which the network weights are optimized for task performance, and "structure
learning", during which neurons are group nodes are optimized with a knowledge distillation-like approach.
The method is evaluated on two five-task datasets (cityscapes, NYUv2) and compared with four state-of-the-art methods, with
favorable results. Generalization and robustness to corruptions are compared to a baseline single-network.

**Summary Of The Review:**

This paper presents a method that performs well, as evidenced by solid experiments, but suffers from an incomplete / unclear description of the methodology. I am unsure regarding the novelty of this work.

---

### Official Review · Reviewer_mKd4 · 2022-11-09

**Confidence:** 4
**Correctness:** 3
**Technical Novelty And Significance:** 3
**Empirical Novelty And Significance:** Not applicable
**Recommendation:** 6

**Clarity, Quality, Novelty And Reproducibility:**

The paper is fairly written and well-organized.
The work seems to be original.
The authors indicated that they would make their code available upon acceptance. Except few minor training details, there should not be any issue of reproducibility.

**Strength And Weaknesses:**

Pros:
+  The paper is well-written and it’s quite well-organized. It was a pleasure reading it.
+ Task-relatedness is extremely crucial for multi-task learning. Grouping tasks based on local task representations and transferring knowledge to a new group layer is very interesting.
+  Exposition of the results is good; particularly the ablation study reporting the effect of alignment, average initialization, and attention based knowledge amalgamation.

Cons:
- The generalization experiment reported in 4.2 is not quite clear to me. Did the authors perform cross-dataset evaluation (trained on CS, tested on NYU, and vice-versa) or cross-task evaluation?
- How would the MTSL work if there are one or multiple tasks requiring varying architectures (e.g., classification from the encoder only and segmentation from the encoder-decoder )?
- What's the reason behind choosing the particular encoder and decoder networks? Why the decoder is changed from DeepLabv3 to ResNet blocks?
-  The authors mentioned that the multitask loss is a weighted sum of all individual task losses. It should be clarified how the task losses are weighted. Which task is to prioritize and when?
- What value did the authors assign to the balancing factor \lambda in eq.(1)?
- Some of the terms are not defined such as, GMac, etc. Table captions could be revised clarifying all the terms. All the performance metrics should be discussed before using them in the tables.
- it looks like MTSL is consistently poorer than One-Net in the case of noise corruption in Table 3. Why so?
- Automatic transitioning between the learning phases would be more appropriate as the model is supposed to learn the architecture as well as its parameters automatically.

**Summary Of The Paper:**

The paper presents a structural learning algorithm for concurrently learning the multi-task learning architecture and its parameters. The multitask learning is performed by the creation and removal of neurons based on local similarity. The proposed method is validated on the Cityscapes and NYUv2 datasets for five different dense prediction tasks. Experimental results include SOTA comparison, generalization and inference efficiency, robustness to natural corruptions, and ablation study.

**Summary Of The Review:**

I found the idea of multi-task structural learning with task similarity and neuron creation/removal quite interesting. The authors presented results from a number of experiments for validating their proposed method. Although the results are not super convincing, I believe there is the true potential of MTSL in learning similar tasks within a single model.

---

### Author Response · Authors · 2022-11-10
**Summary of rebuttal revisions**

To facilitate the tracking of changes in the revised paper based on suggestions by the reviewers, we provide a summary of revisions.

* Section E visualizes and infers the converged architectures of LTB, BMTAS, and MTSL. We show that MTSL leads to more sharing among tasks and improved inference efficiency.
* Section H presents the robustness to natural corruptions of all state-of-the-art methods along with MTSL. We show that MTSL is more robust to weather, digital, and blur corruptions.
* We ran more experiments to test the sensitivity of BMTAS to the weight of resource loss. Section G provides this sensitivity analysis.
* Section F details CKA and the ability to optimize CKA.
* Clarity improvement in the abstract to state when MTSL moves on to upcoming layers.
* Improved related works by incorporating a recent work.
* Improved interpretation and fixes of Equations 1 and 2 to remove ambiguity.
* Clarity on the initialization of MTSL and compared works. All results including that of MTSL are presented by initializing ResNet with ImageNet pretrained weights and the rest of the weights randomly.
* Specified additional hyperparameters such as CKA regularization loss weight, and weights of task losses.
* Improved clarity by adding missing definitions of terms such as GMac.
* Improved interpretation of results in section 4.1 detailing cross-stitch, LTB-R, BMTAS-R, and MTSL-R.
* Made an inference in section 4.2 concerning attribution to brain-related aspects as a hypothesis.

---

### Author Response · Authors · 2022-11-17
**General comments to reviewers**

As the deadline for discussion period is approaching, we would like to request the reviewers to let us know whether the rebuttal sufficiently addresses the raised concerns. If there are further concerns, we would be glad to address them as quickly as possible. We are looking forward to the response from the reviewers.

---

### Decision · Program_Chairs · 2023-01-20

**Decision:**

Reject

**Justification For Why Not Higher Score:**

Empirical evaluation of the method revealed marginal performance.

**Justification For Why Not Lower Score:**

N/A

**Metareview: Summary, Strengths And Weaknesses:**

The paper proposes a new method for multi-task learning involving the learning of the architecture of the network as well as its weights. While reviewers agreed that the method is novel, they were not convinced that the method improved significantly upon SOTA. Further, they were unconvinced about the biological motivation for the method.

**Summary Of Ac-Reviewer Meeting:**

At the video conference, the reviewers were all in complete agreement that the method was novel, but also that the empirical evaluation was unconvincing. They encouraged the authors to help the reader understand why their method constitutes a significant advance. The sole reviewer advocating for borderline acceptance also changed their mind, agreeing with the other reviewers for rejection.